# FV-100 for the Treatment of Varicella-Virus (VZV) Infections: Quo Vadis?

**DOI:** 10.3390/v14040770

**Published:** 2022-04-07

**Authors:** Erik De Clercq

**Affiliations:** Department of Microbiology, Immunology and Transplantation, Rega Institute for Medical Research, KU Leuven, 3000 Leuven, Belgium; erik.declercq@kuleuven.be

**Keywords:** Cf1743, FV-100, varicella-zoster virus, herpes zoster, thymidine kinase, DNA polymerase

## Abstract

The bicyclic nucleoside analogue (BCNA) Cf1743 and its orally bioavailable prodrug FV-100 have unique potential as varicella-zoster virus (VZV) inhibitors to treat herpes zoster (shingles) and the therewith associated pain, including post-herpetic neuralgia (PHN). The anti-VZV activity of Cf1743 depends on a specific phosphorylation by the VZV-encoded thymidine kinase (TK). The target of antiviral action is assumed to be the viral DNA polymerase (or DNA synthesis in the virus-infected cells).

## 1. Introduction

In 2010, Marco Migliore stated in *Antiviral Chemistry and Chemotherapy* that FV-100, the hydrochloride (named hydrochloric) salt of the 5′-valyl ester of Cf1743, was the most potent and selective anti-varicella-zoster virus (VZV) agent reported to date [1]. As of today, in 2022, this statement still holds true. The bicyclic furo [2,3-d] pyrimidine nucleoside analogue (BCNA) Cf1743 and its 5′-valyl ester FV-100 have been the subject of a number of review articles by myself [2,3,4,5,6,7] and my colleagues, Graciela Andrei and Robert Snoeck [8,9,10], and other authors [11,12,13], but neither Cf1743 nor FV-100 or any derivative thereof have so far been licensed for clinical use in the therapy of VZV infections, such as herpes zoster. Here, I will address the current state of the art of FV-100 and evaluate the issues that still remain to be resolved.

## 2. History

The original BCNAs were obtained as by-products in Pd-catalyzed coupling of terminal alkynes with 5-iodo-nucleoside analogues, resulting in the preparation of 5-alkynyl-2′-deoxyuridines starting from 5-iodo-2′-deoxyuridine [14]. BCNAs bearing an aryl side chain were then prepared [15], from which originated Cf1743 (Figure 1).

## 3. Anti-VZV Activity

In human embryonic lung (HEL) fibroblast cell cultures, Cf1743 proved clearly more potent against a wide range of clinical VZV isolates and reference laboratory VZV strains than BVDU, acyclovir (ACV), penciclovir (PCV) and foscarnet (PFA) (Figure 2) [3,16]. Cf1743 was not active against ACV- and BVDU-resistant VZV strains bearing a mutation in the viral thymidine kinase gene but kept their inhibitory potential against VZV strains with mutations in the VZV DNA polymerase gene [16]. Mutant virus strains selected in the presence of the BCNAs were solely cross-resistant to drugs such as ACV and BVDU that depend on metabolic activation by the viral thymidine kinase for their antiviral action [16].

## 4. Metabolism (Anabolism, Catabolism)

The metabolism of the BCNAs, as exemplified for CF1743, is, as far as anabolism is concerned, mainly comparable to that of BVDU (Figure 3) in that it entirely depends on its successive phosphorylation to the 5′-monophosphate (BCNA-MP) and 5′-diphosphate (BCNA-DP). The latter is then assumed to be converted to the 5′-triphosphate (BCNA-TP) by a nucleoside diphosphate kinase (NDPK) [3]. The target for the antiviral action is postulated to be the viral and/or cellular DNA polymerase, but this has not been directly demonstrated.

While BVDU [(*E*)-5-(2-bromovinyl)-2′-deoxyuridine] (BVDU) is promptly recognized by thymidine phosphorylase to be converted to its free base [(*E*)-5-(2-bromovinyl)uracil] (BVU), this conversion does not occur with the BCNAs, i.e., Cf1743 [17]. Accordingly, BCNAs cannot give rise to the formation of any products that, like BVU, might inhibit the activity of dihydropyrimidine dehydrogenase (DPD), and, consequently, interfere with the degradation of pyrimidine analogues such as 5-fluorouracil (FU). Due to the increased plasma FU levels, the concomitant use of FU with the BVU generating anti-VZV compound (*E*)-5-(2-bromovinyl)-1-β-D-arabinosyluracil (BVaraU, sorivudine) has led to some casualties of Japanese patients [17]. This problem can obviously be averted by using BCNAs such as Cf1743 not interfering with FU metabolism.

In contrast with BVDU, which is specifically active against both herpes simplex virus type 1 (HSV-1) and VZV, the antiviral activity spectrum of CF1743 is confined to VZV. The VZV-encoded thymidine kinase (TK) [18] successively phosphorylates the BCNA Cf1743 to its 5′-mono- and 5′-diphosphate (Figure 3). What happens next in this process has not been elucidated. It is presumed that the BCNA-DP is subsequently converted to its 3′-triphosphate (BCNA-TP) by a nucleoside diphosphate kinase (NDPK). The BCNA-TP is assumed to be the active metabolite of CF1743, interacting with the viral DNA synthesis (or DNA synthesis within VZV-infected cells).

## 5. Target of Antiviral Action

Whether CF1743 would eventually be targeted at the DNA synthesis of the VZV-infected cells has not been directly demonstrated. It could be hypothesized that if incorporated into the DNA, CF1743 may form two hydrogen bonds with guanine (Figure 4), thus leading to base pairing with guanine. This highly speculative hypothesis may pertain to the final antiviral action of CF1743 and remains entirely open for further experimental exploration.

## 6. Switch from Cf1743 to FV-100

Cf1743 was found to exhibit poor oral bioavailability (~14%) [19]; therefore, its 5′-valyl ester was designed, which as its HCI salt emerged as the most promising prodrug of CF1743 and was designated as FV-100. There was a significant (~10-fold) boost in both AUC and C_max_ for FV-100 over Cf1743 (Figure 5) [1]. Accordingly, FV-100 was chosen as the clinical BCNA candidate for the treatment of herpes zoster (shingles) [19].

## 7. Clinical Evaluation

Three randomized double-blind, placebo-controlled clinical trials were conducted [20]. The overall conclusion of these clinical studies was that the pharmacokinetic and safety profiles of FV-100 justified further investigation for the treatment of herpes zoster and the prevention of postherpetic neuralgia (PHN) with once-daily oral dosing, i.e., 400 mg QD for 7 days, without modifications for the elderly or renally impaired patients [20].

The potential of FV-100 for the prevention of PHN and the treatment of acute herpes zoster-associated pain has been (again) emphasized in a randomized-controlled clinical trial [21]. The data presented by Tyring et al. point to the potential role for FV-100 in the reduction in subacute and chronic pain as well as the prevention of PHN. The safety profile of FV-100 remains favorable both in isolation and when compared to valacyclovir. Current antiviral medications have limited effectiveness in the reduction in subacute and chronic pain; they do not satisfactorily prevent or adequately treat PHN and require dosing modifications in patients with renal insufficiencies. The efficacy results from this study support further investigation of FV-100 to address these unmet medical needs [21]. 

## 8. Conclusions and Perspectives

FV-100 has been hailed [1] as the most potent and selective anti-VZV agent reported to date (2010). Yet, neither FV-100 nor its active ingredient, Cf1743 (Figure 1), are currently on the market as antiviral drugs. What is hampering their clinical development?

Of crucial importance is that the target of antiviral action for CF1743, i.e., the VZV DNA polymerase has been presumed but not unequivocally demonstrated. Additionally, direct evidence for the final conversion of the Cf1743 diphosphate to the triphosphate (presumably by the nucleoside diphosphate kinase (NDPK)) is still lacking.

That the esterification of Cf1743 by valine greatly increases its oral bioavailability is reminiscent of what is known for valacyclovir versus acyclovir) and valganciclovir versus ganciclovir), but how would other aminoacyl esters behave in this regard?

The major discomfort resulting from herpes zoster (shingles) is acute zoster-associated pain and post-herpetic neuralgia (PHN), but how would the efficiency of Cf1743 (or FV-100) compare in this respect with the efficiency of established treatments such as valacyclovir, famciclovir and BVDU? The topical delivery of Cf1743 was claimed to limit the extent and severity of PHN [22], but the topical treatment of herpes zoster is not routinely advocated in medical practice.

The conversion of CF1743 to phosphoramidate protides would seem most attractive to honor Prof. Chris McGuigan’s legacy, and deserves more attention, certainly in attempts to enhance oral bioavailability. For 2′-fluro derivatives of Cf1743 [23,24], the ProTide approach was determined to be not successful [23], but this conclusion was based solely on “not successful in bypassing VZV TK deficiency”. As demonstrated for tenofovir alafenamide [25], conjugation with a phosphoramidate moiety may greatly enhance the oral bioavailability.

News provided by Contravir Pharmaceuticals, Inc., on 03 March 2016 dealt with positive results confirming the safety of its shingles candidate FV-100 in a drug–drug interaction study [26]. The comparative study of FV-100 versus valacyclovir for the prevention of post-herpetic neuralgia was further documented in a report from the U.S. National Library of Medicine (ClinicalTrials.gov [27]). Contravir Pharmaceuticals then announced, on 22 July 2019, that their name had changed to Hepion Pharmaceuticals, Inc. [28]. None of these reports [26,27,28] mentioned any adverse effects encountered with FV-100.

## 9. Epilogue

This article does not attempt to provide a possible comparative evaluation of the market value of vaccination versus therapy in the management of varicella-zoster virus (VZV) infection. The aim of this article is primarily intended to assess the benefits of FV-100 in the treatment of VZV infections. By definition, it does not want to assess the potential benefit of any vaccination, because the latter approach is based on the prevention (prophylaxis) of VZV infections, whereas with FV-100 I wanted to highlight its potential in the therapy of VZV infections, including herpes zoster and the clinical symptoms (i.e., PHN) thereof.

## Figures and Tables

**Figure 1 viruses-14-00770-f001:**
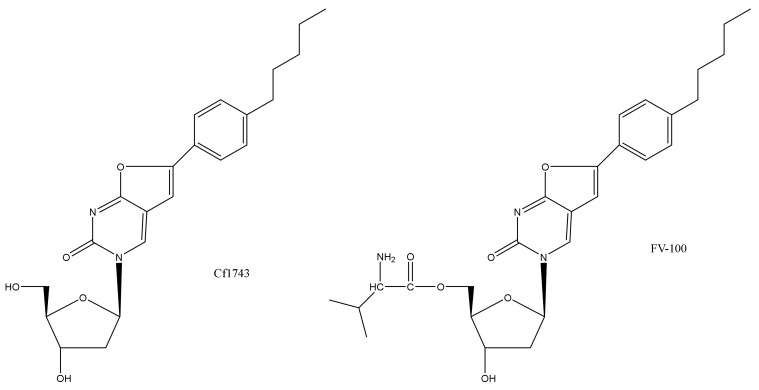
Chemical structures of Cf1743 and FV-100.

**Figure 2 viruses-14-00770-f002:**
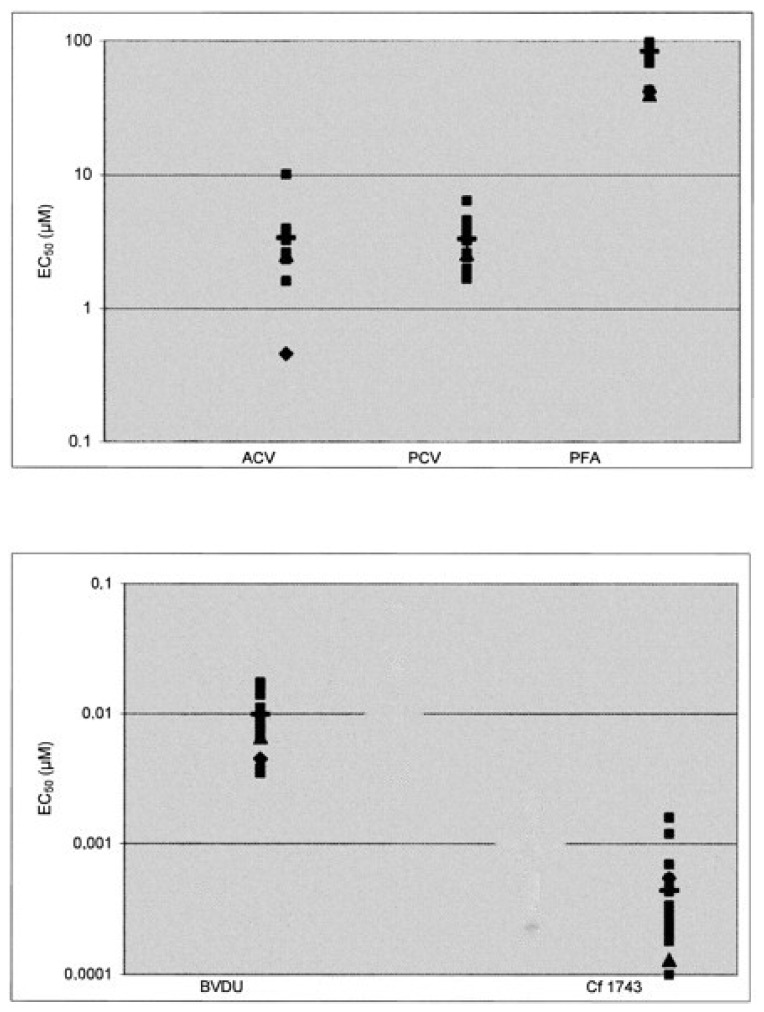
Activity of CF1743 as compared to that of acyclovir (ACV), penciclovir (PCV) and foscarnet (phosphonoformic acid, PFA) and brivudine (BVDU) against clinical VZV isolates (■) and VZV reference strains Oka (▲) and YS (♦). Mean values for clinical isolates are indicated by horizontal lines (▬). Data taken from Refs. [3,16].

**Figure 3 viruses-14-00770-f003:**
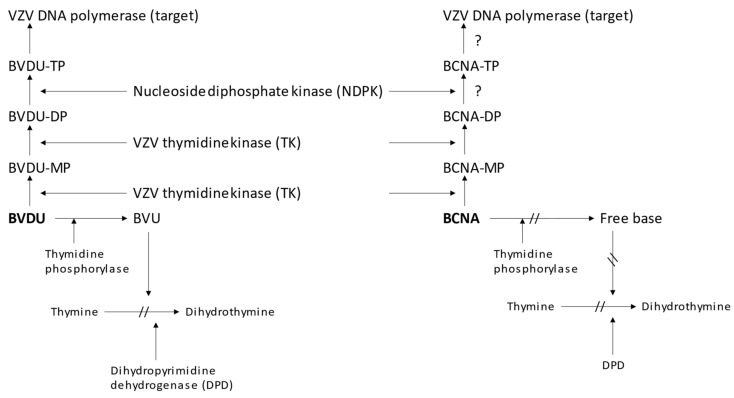
Anabolic and catabolic steps involved in the metabolism of BVDU and BCNA.

**Figure 4 viruses-14-00770-f004:**
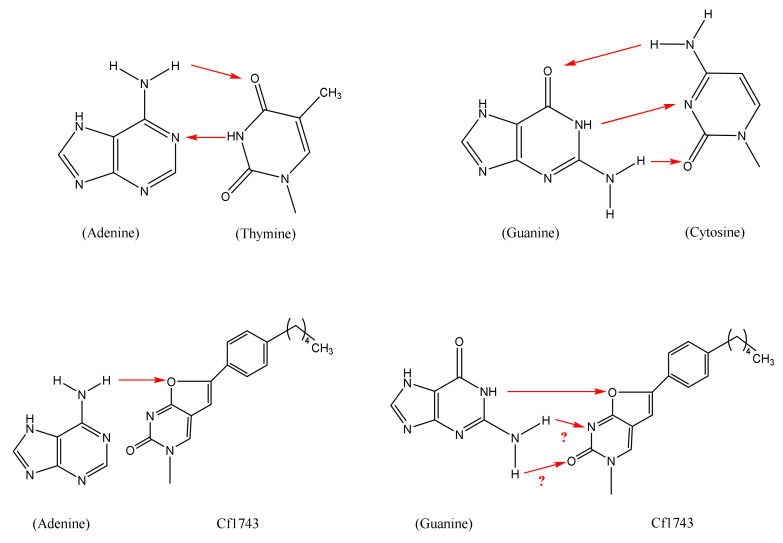
Hypothesis: Watson–Crick type of base pairing (based on formation of two hydrogen bonds) not possible between adenine and Cf1743, but possible between guanine and Cf1743.

**Figure 5 viruses-14-00770-f005:**
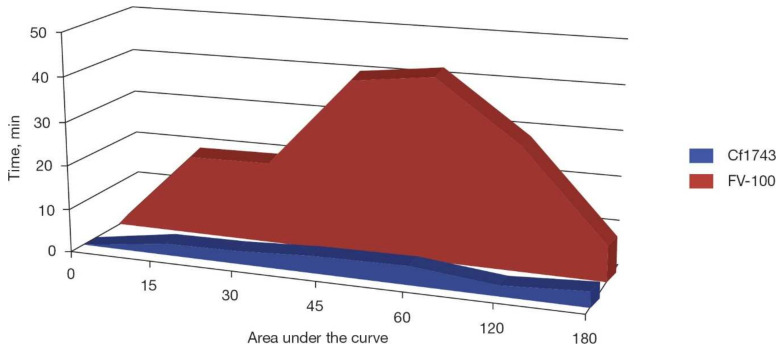
Oral bioavailability of FV-100 as compared to Cf1743 as monitored by area under the curve (AUC) versus time. (Figure from Ref. [1]).

## Data Availability

Not applicable.

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
