# Peer review of "FV-100 for the Treatment of Varicella-Virus (VZV) Infections: Quo Vadis?"

_viruses, 2022, doi:10.3390/v14040770_

Round 1

Reviewer 1 Report

The authors have reviewed an orally available nucleoside analogue drug with antiviral activity (FV -100) for the possible treatment option of shingles. 

As of now, many antiviral therapies are available for the management of shingles. 

Apparently, the manuscript is poorly written and not suitable for publication in Viruses Journal (MDPI) for several reasons: first, the study purpose was not made clear. Many similar studies are available in the literature, however, the authors failed to cite some of them, especially some key references. Thus the research gap was not made clear. Second, no obvious flow has been observed and quite a lot of details were missing. Third, results are within expectation as nothing really new was found compared to the literature. Fourth, the discussion is very superficial as the authors did not discuss how their study results compared to others in the literature. Thus, it is difficult to see its value or contribution. Finally, the conclusion needs to be addressed in concise version and clearly demonstrate the output.     Specific comments: 1. Language editing is necessary as there are many grammatical errors throughout the manuscript. 2. Abbreviations should be spelled out with definitions when they first appear in the text. 3. Limited references for, especially with increasing publications. 4. some part of the Title of the study needs to convert into the English language.  5. In the introduction part, no need to write authors name. and it's advisable to use We, instead of I. 

Author Response

Extensive editing of English language and style required
This must have been a comment for another article

Apparently, the manuscript is poorly written, etc…

  • “Many grammatical errors” If so, can you please indicate which ones?
  • “Title of the study needs to be converted into the English language” Apparently the Reviewer does not understand Quo Vadis? (= where are you going?)

The comments of Reviewer 1 are full of inconsistencies and contradictions which cannot be remedied.

Reviewer 2 Report

The modified manuscript was well-modified and the contents was well-written. 

In figure 2, it is difficult to understand the position of the horizontal axis and the meaning of the data in the table, so the arrangement should be modified to make it easier to understand.

Author Response

The modified manuscript was well modified…
I do not understand as this manuscript was not a modified manuscript.

In Figure 2, it is difficult to understand the position of the horizontal axis…
Figure 2 has been modified to comply with the Reviewer’s remark.

Reviewer 3 Report

In this manuscript the author reviews the current state of FV-100 research against VZV and related herpes viruses. The information is presented in an easy to read and straight forward manner. 

Author Response

The English used is correct and readable
The information is presented in an easy to read and straightforward manner
In contrast with Reviewer 1’s comments which are ill-inspired, the comments of Reviewer 3 are re-assuring. To this Reviewer: thank you so much!

Round 2

Reviewer 1 Report

The authors have answered all of my queries. 

Author Response

Thank you for your comments.